# PROBABILISTIC KNOWLEDGE GRAPH EMBEDDINGS

## ABSTRACT

We develop a probabilistic extension of state-of-the-art embedding models for link prediction in relational knowledge graphs. Knowledge graphs are collections of relational facts, where each fact states that a certain relation holds between two entities, such as people, places, or objects. We argue that knowledge graphs should be treated within a Bayesian framework because even large knowledge graphs typically contain only few facts per entity, leading effectively to a small data problem where parameter uncertainty matters. We introduce a probabilistic reinterpretation of the DistMult (Yang et al., 2015) and ComplEx (Trouillon et al., 2016) models and employ variational inference to estimate a lower bound on the marginal likelihood of the data. We find that the main benefit of the Bayesian approach is that it allows for efficient, gradient based optimization over hyperparameters, which would lead to divergences in a non-Bayesian treatment. Models with such learned hyperparameters improve over the state-of-the-art by a significant margin, as we demonstrate on several benchmarks.

## 1 INTRODUCTION

In 2012, Google announced that it improved the quality of its search engines significantly by utilizing knowledge graphs (Eder, 2012). A knowledge graph is a dataset of facts represented in terms of triplets (*head, relation, tail*), where *head* and *tail* represent entities in the world, and a *relation* is a property that describes the relationship between the two entities. As an example, when the *head* entity is 'Paris', and the *relation* is 'is in', then the *tail* entity could be 'France' or 'Europe'.

While the number of possible relations among entities is enormous, the amount of edges in empirical knowledge graphs is often rather small. It is therefore desirable to complete the missing edges in a knowledge graph algorithmically based on patterns detected in the observed part of he graph (Nickel et al., 2016a). Such link prediction has become an important subfield of artificial intelligence (Bordes et al., 2013; Wang et al., 2014; Lin et al., 2015; Nickel et al., 2016c; Trouillon et al., 2016; Wang & Li, 2016; Ji et al., 2016; Shen et al., 2016; Xiao et al., 2017; Shi & Weninger, 2017; Lacroix et al., 2018).

In statistical relational learning, link prediction is done by inferring/learning explicit *rules* about relationships and then applying these inferred rules in order to reason about unobserved facts (Friedman et al., 1999; Kersting et al., 2011; Niu et al., 2012; Pujara et al., 2015). A complementary, highly scalable approach relies on embedding models (Kadlec et al., 2017; Nguyen, 2017). These models represent entities and relationships in terms of low-dimensional semantic vectors and implicitly encode the relationships in the knowledge graphs in terms of three-way mathematical operations among these vector triplets (Yang et al., 2015; Trouillon et al., 2016). In the early works, different embeddings were used for the two tasks of head and tail prediction. Lacroix et al. (2018) proposed to map these two tasks to just the tail prediction task by augmenting the data by reciprocal facts with inversed relations. In the present work, we also use this data augmentation technique to pave the way for a probabilistic interpretation of knowledge graph embedding models. The probabilistic formulation allows us to keep track of parameter uncertainty by applying Bayesian methods.

We argue that the parameter uncertainty in knowledge graph embedding models can be significant, calling for a Bayesian approach. As the number of parameters grows linearly with the number of entities, and every entity is only connected to a small number of other entities by observed links, this effectively leads to a small data problem, even if the knowledge graph as a whole may be huge. For example, Figure 1 shows that for a popular large-scale knowledge base, about $80\%$ of entities have

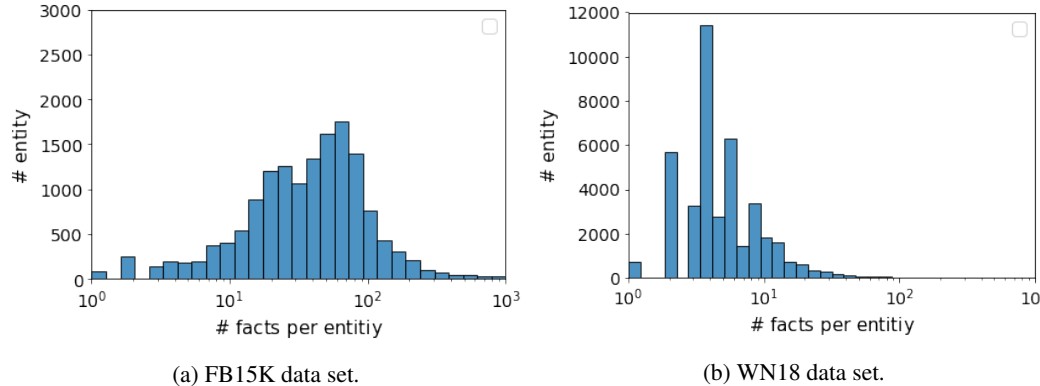

(a) FB15K data set.            (b) WN18 data set.

Figure 1: Histograms of the number of facts per entity in the training set of two data sets. Note the log scale of the horizontal axis. The plots show that the distributions are left-skewed. For example, in the FB15K dataset, a third of the entities have less than 25 facts in the whole training set, justifying the claim that even large knowledge bases involve small-data problems.

less than 80 observed facts. Similar observation have been made for recommender systems, where most users interact only with a small subset of the items (Gopalan et al., 2014; 2015; Liang et al., 2018). As in recommender systems, we show that a Bayesian approach can be highly beneficial in knowledge graph embeddings. This paper presents such a Bayesian formulation and presents an interpretation of popular embedding models as generative models of relational facts.

The benefits of a Bayesian formulation of knowledge graph embeddings are two-fold. First, this formulation allows us to infer posterior distributions of embedding vectors, giving us uncertainty estimates. An arguably more important aspect is that of hyperparameter tuning, as the state-of-the-art embedding models employ a large number of hyperparameters (regularizers), one for every entity (Lacroix et al., 2018). Kadlec et al. (2017) showed that careful hyperparameter tuning can dramatically increase predictive performance. While there is no classical way to scalably tune hyperparameters on a per-entity level without some heuristic, our Bayesian approach allows for gradient-based hyperparameter optimization on the marginal likelihood of the data (Bishop, 2006). This approach, termed variational expectation maximization (variational EM (Bernardo et al., 2003)), still works if variational approximations are used (Jordan et al., 1999; Blei et al., 2017; Zhang et al., 2017).

We propose a simple, principled, and efficient way to tune hyperparameters using variational EM. This approach lower-bounds the marginal likelihood of the data and allows this bound to be optimized for hyperparameters without running into degenerate solutions (Bernardo et al., 2003; Mandt et al., 2017). This is in contrast to point estimation, which results in degeneracies when jointly optimizing over model parameters and hyperparameters. As a result, our approach learns a macroscopic number of hyperparameters, resulting in a new state-of-the-art performance on link prediction.

Our contribution in this work can be summarized as follows:

- We present a probabilistic interpretation of existing knowledge graph embedding models such as ComplEx and DistMult. We reformulate these models as generative models for relational facts, which paves the way for Bayesian inference in these models.
- We apply stochastic variational inference to scalably estimate an approximate posterior for each entity and relation embedding in the knowledge graph. This approach not only allows us to estimate uncertainty, but more importantly allows for gradient-based hyperparameter optimization by stochastic gradient descent on the optimized variational bound (variational EM). Implementing our methodology in existing models requires only a few lines of additional code, and it does not increase the computational complexity compared to a non-Bayesian training approach.
- We show experimentally that our method reaches new state-of-the-art results in link prediction. We present results on several large data sets using two popular models which we trained using variational EM. We observe that our improvements are most notable for entites with few training points, as we show by introducing a new, balanced evaluation metric.

The paper is structured as follows. In Section 2, we review the related work about knowledge graph embeddings and variational inference. In Section 3, we present the generative model for relational facts and reinterpret existing loss functions as the maximum a posterior estimator of the generative

model. In Section 4, we propose our approximate Bayesian expectation maximization algorithm for finding the hyperparameters. Finally, in Section 5 we test the hyperparameter learning algorithm for the link prediction task on two different embedding algorithms using standard performance metrics.

## 2   RELATED WORK

Related work to this paper can be grouped into link prediction algorithms and variational inference.

**Link Prediction.**   Knowledge graphs and link prediction gained a lot of attention; for a review see (Nickel et al., 2016b). Most link prediction algorithms use latent features of entities or relationships, also termed embeddings. TransE proposed by Bordes et al. (2013) is one example, where each entity and relation are embedded in a $K$-dimensional space. The embeddings are found such that for a given fact $(h, r, t)$, the sum of embeddings of $h$ plus $r$ be close to $t$, and far from other entities.

Many models are based on tensor factorization, where an adjacency tensor is built from the triplet facts, and entities and relations are embedded to a $K$-dimensional space. Different methods differ in the structure of the embedded vectors. Hitchcock (1927) assume that an entity's embedding depends on whether the entity appears as the head or tail of a fact. DistMult simplified the model by assuming that tails and heads have the same embeddings (Yang et al., 2015; Toutanova & Chen, 2015). ComplEx extended DistMult by embedding the entities and relations to a complex space to allow for nonsymmetric relations. Lacroix et al. (2018) introduced reciprocal relations and trained with the full log-loss instead of binary loss with negative sampling, which improved the results.

Link prediction algorithms are susceptible to overfitting, and the importance of hyperparameter choices has been pointed out by several authors, e.g., (Kadlec et al., 2017; Lacroix et al., 2018). Different reguralizers have been proposed, the most standard one being the 2-norm (Yang et al., 2015). A heuristic that often produces the best results (e.g., in (Lacroix et al., 2018)) is to choose the strength of the regularizer proportional to the frequency of the entities (see e.g., (Srebro & Salakhutdinov, 2010)). In contrast, our approach learns entity-dependend hyperparameters without relying on heuristics. Also, Lacroix et al. (2018) showed that the 2-norm is not a valid tensor norm, and that the 3-norm is more appropriate for triplet data.

**Variational Inference.**   Variational inference (VI) is a powerful technique to approximate a Bayesian posterior over latent variables given observations (Jordan et al., 1999; Blei et al., 2017; Zhang et al., 2017). Besides approximating the posterior, VI also estimates the marginal likelihood of the data. This allows for gradient-based hyperparameter tuning, a technique called variational EM (Bernardo et al., 2003), which is the main benefit of the Bayesian approach used in this paper.

Our paper builds on recent probabilistic extensions of embedding models to Bayesian models, such as word (Barkan, 2017) or paragraph (Ji et al., 2017) embeddings. In these works, the words are embedded into a $K$-dimensional space. It has been shown that using a probabilistic approach leads to better performance on smaller data, and allows these models to be combined with powerful priors, such as for time series modeling (Bamler & Mandt, 2017; Jähnichen et al., 2018). Yet, the underlying probabilistic models in these papers are very different from the ones considered in our work.

## 3   KNOWLEDGE GRAPH EMBEDDINGS AS GENERATIVE MODELS OF FACTS

In this section, we describe a generative model for relational facts. Then, we show that existing tensor factorization models can be interpreted as a maximum a posteriori (MAP) estimation of the parameters of the generative model (Bishop, 2006). In Section 4, we move to our main contribution. We use our generative model to derive an approximate Bayesian inference algorithm to optimize over hyperparameters (regularizers) in a principled way.

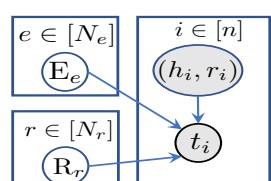

Figure 2: The generative process of the triples $(h, r, t)$.

Let $\mathbb{S} = \{(h_i, r_i, t_i)\}_{i=1,\ldots,n}$ be the set of $n$ triplet facts, with heads $h_i \in [N_e]$, tails $t_i \in [N_e]$, and relations $r_i \in [N_r]$. Here, $N_e$ and $N_r$ are the number of distinct entities and relations that we model. From now on, we use the data augmentation technique used in (Lacroix et al., 2018) (see Remark 2 in

the appendix). It reduces the task of performing both head and tail predictions to that of performing only tail predictions by introducing reciprocal relations. Thus, given a head $h$ and relation $r$, the link prediction boils down to finding the tail $t$ in the fact $(h, r, t)$.

For simplicity, we present the generative process at the example of the DistMult model (Yang et al., 2015). Generalization to other models, such as ComplEx (Trouillon et al., 2016) or CP (Hitchcock, 1927) is straight-forward. In DistMult, each entity and relation is embedded in the latent vector space $\mathbb{R}^K$. Let $\mathbf{E} \in \mathbb{R}^{N_e \times K}$ (similarly $\mathbf{R} \in \mathbb{R}^{N_r \times K}$) be the embedding matrix for entities (relations), where $E_e \in \mathbb{R}^K$ ($R_r \in \mathbb{R}^K$) is the embedding vector for entity $e \in [N_e]$ (relation $r \in [N_r]$).

**The Score.** Embedding models such as DistMult rely on a score $X_{h,r,t}$ which plays a role both in the probabilistic and classical formulation of the model. This score assigns a scalar value to each combination of the embeddings $E_h$, $R_r$, and $E_t$. In DistMult, the score is $X_{h,r,t} = \sum_{i=1}^{K} E_{hi} R_{ri} E_{ti}$. In the conventional approach, the embedding vectors are trained in such a way that observed triplets have high scores. As we show below, the score also enters the probabilistic extension of the model.

**Generative Process.** We can now present our perspective of knowledge graph embeddings as generative models. Figure 2 shows a graphical model. The generative process is as follows:

- For each entity $e \in [N_e]$, draw an embedding $E_e \in \mathbb{R}^K$ from a prior $p(E_e|\lambda_e)$, e.g., a normal distribution. Similarly, for each relation $r \in [N_r]$, draw an embedding $R_r \in \mathbb{R}^K$ from a prior $p(R_r|\lambda_r)$. Here, $\lambda_e$ and $\lambda_r$ are hyperparameters.

- Repeat for each triplet index $i \in \{1, \ldots, n\}$:
    - Draw a head $h_i$ and a relation $r_i$ from a discrete joint distribution $P(h_i, r_i)$.
    - Draw a tail $t_i \sim \text{Multinomial}(\text{softmax}_t(X_{h_i, r_i, t}))$.

Above, the softmax normalizes the scores over the tail index, $\text{softmax}_t(X_{h,r,t}) = \frac{\exp X_{h,r,t}}{\sum_{t'} \exp X_{h,r,t'}}$. We see that the resulting generative model produces fact triplets conditioned on semantic embeddings of entities and relations. This concludes the generative process perspective.

**MAP Estimation Recovers Original Model.** The cost function used for learning $\mathbf{E}$ and $\mathbf{R}$ in the existing works can be seen as maximum a posteriori (MAP) estimation of the generative model above, where we optimize the log joint distribution over the embeddings $\mathbf{E}$ and $\mathbf{R}$:

$$\mathbf{E}^\star, \mathbf{R}^\star = \arg\max_{\mathbf{E}, \mathbf{R}} \log p(\mathbf{E}, \mathbf{R}, \mathbb{S}|\lambda) = \arg\max_{\mathbf{E}, \mathbf{R}} \left\{ \log p(\mathbb{S}|\mathbf{E}, \mathbf{R}) + \log p(\mathbf{E}, \mathbf{R}|\lambda) \right\}. \quad (1)$$

In the DistMult model, the loss function with a Gaussian prior with precision $\lambda$ becomes

$$L(\mathbf{E}, \mathbf{R}, \lambda) = -\log p(\mathbb{S}|\mathbf{E}, \mathbf{R}) - \log p(\mathbf{E}, \mathbf{R}|\lambda), \quad (2)$$

where the negative log likelihood is just the cross entropy,

$$-\log p(\mathbb{S}|\mathbf{E}, \mathbf{R}) = \sum_{(h,r,t) \in \mathbb{S}} \left( -X_{h,r,t} + \log \sum_{t'} \exp X_{h,r,t'} \right) + c, \quad (3)$$

and the negative log prior leads to the regularizer

$$-\log p(\mathbf{E}, \mathbf{R}|\lambda) = \frac{1}{2} \sum_{e \in [N_e]} \left( \lambda_e \|E_e\|^2 - K \log(\lambda_e) \right) + \frac{1}{2} \sum_{r \in [N_r]} \left( \lambda_r \|R_r\|^2 - K \log(\lambda_r) \right) + c. \quad (4)$$

Up to the terms $K \log(\lambda_{e/r})$ in Eq. 4, which do not affect the minimization over $\mathbf{E}$ and $\mathbf{R}$, Eqs. 2-4 recover precisely the loss function used in (Lacroix et al., 2018).

## 4 GRADIENT-BASED HYPERPARAMETER TUNING VIA VARIATIONAL EM

In this section, we describe an efficient method to learn optimized hyperparameters $\lambda_e$ and $\lambda_r$ for each entity and each relation embedding in the model. We stress that such optimization could not be achieved via classical grid search due to the sheer number of hyperparameters; this number would

have to be reduced by some heuristic (Srebro & Salakhutdinov, 2010). Our method is based on optimizing the marginal likelihood over $\lambda_e$ and $\lambda_r$ using variational inference.

Recall that in a knowledge graph, the number of triplets per entity is typically small (see Figure 1). Therefore, the cross entropy (log likelihood, Eq. 3) contains only few terms per entity, and thus the regularizer (log prior, Eq. 4) has a strong influence on the optimization.

Finding good hyperparameters $\lambda_e$ and $\lambda_r$ for each entity $e$ and relation $r$ would usually involve expensive cross validations and grid search over many parameters. Simply minimizing the loss $L$ in Eqs. 2-4 over both hyperparameters $\lambda_{e/r}$ and model parameters $\mathbf{E}$ and $\mathbf{R}$ does not work because $L$ is not bounded from below. Setting $\mathbf{E}$ and $\mathbf{R}$ to zero and $\lambda_e, \lambda_r \to \infty$ would send $L \to -\infty$ due to the terms $-\frac{K}{2}\log(\lambda_{e/r})$ in Eq. 4. Note that ignoring these terms would not solve the problem, as the optimization would then set $\lambda_e$ and $\lambda_r$ to zero and the model would become unregularized.

---

**Algorithm 1** Gradient Based Tuning of Hyperparameters $\lambda$

---

1: **Input:** Number of training steps $T$; number of initialization steps $T_0$
2: **Initialize:** $\lambda$ and $\gamma = (\mu, \sigma)$
3: **Pre-train the model using SGD:** $\mu^E, \mu^R = \arg\min_{\mathbf{E}, \mathbf{R}} L(\mathbf{E}, \mathbf{R}, \lambda)$
4: **for** $t = 0 : T$ **do**           ▷ Variational EM update for $T$ iterations
5:    draw mini-batch $\mathcal{B} \in \mathbb{S}$
6:    draw uniform noise samples, $\epsilon_e \sim N(0, I)$ and $\epsilon_r \sim N(0, I)$
7:    $\gamma \leftarrow \gamma - \nabla_\gamma \left[ L\left(\mu^E + \epsilon^E\sigma^E, \mu^R + \epsilon^R\sigma^R, \lambda\right) - \log\sigma^E - \log\sigma^R \right]$   ▷ $L$ = loss, see Eq. 2
8:    **if** $t \geq T_0$ **then** $\lambda \leftarrow \arg\max_\lambda \mathbb{E}_{q_\gamma}\left[\log p\left(\mathbf{E}, \mathbf{R}|\lambda\right)\right]$    ▷ See appendix for analytic solution
9:    **else**   do not update hyperparameters $\lambda$
10: **Re-train model with learned hyperparameters:** $\mathbf{E}^*, \mathbf{R}^* = \arg\min_{\mathbf{E}, \mathbf{R}} L(\mathbf{E}, \mathbf{R}, \lambda)$

---

**Variational Expectation Maximization.** A well-known approach to avoid degenerate solutions in gradient-based hyperparameter optimization is the expectation-maximization (EM) algorithm (Dempster et al., 1977). This algorithm optimizes the marginal likelihood of a model with hidden variables over hyperparameters by alternating between a gradient step on hyperparameters, and a step in which the hidden variables are integrated out. We use a version of EM based on variational inference, termed variational EM (Bernardo et al., 2003) that avoids the integration step.

The measure of how well a probabilistic model fits the data $\mathbb{S}$ is the marginal likelihood,

$$p(\mathbb{S}|\lambda) = \int p(\mathbf{E}, \mathbf{R}, \mathbb{S}|\lambda)\, d\mathbf{E}\, d\mathbf{R}. \tag{5}$$

We want to find hyperparameters $\lambda$ that maximize the marginal likelihood $p(\mathbb{S}|\lambda)$; however, since the integral in Eq. 5 is intractable, the marginal likelihood is unavailable in closed form. To circumvent this problem, we use variational inference (VI) (Jordan et al., 1999). In VI, one first chooses a family of variational distributions $q_\gamma(\mathbf{E}, \mathbf{R})$, which is parameterized by so-called variational parameters $\gamma$. Evoking Jensen's inequality, the marginal likelihood is then lower-bounded by the *evidence lower bound* (Blei et al., 2017; Zhang et al., 2017), or ELBO, as

$$\log p(\mathbb{S}|\lambda) \geq \mathbb{E}_{\mathbf{E}, \mathbf{R} \sim q_\gamma}\left[\log p(\mathbf{E}, \mathbf{R}, \mathbb{S}|\lambda) - \log q_\gamma(\mathbf{E}, \mathbf{R})\right] =: \text{ELBO}(\lambda, \gamma). \tag{6}$$

The bound is tight if the variational distribution $q_\gamma$ is the true posterior of the model for given $\lambda$. Therefore, maximizing the ELBO over the variational parameters $\gamma$ closes the gap between ELBO and true marginal likelihood, and the ELBO can be taken as a proxy for the latter.

We use the ELBO as a proxy for the marginal likelihood $p(\mathbb{S}|\lambda)$, and we maximize it concurrently over both $\gamma$ and $\lambda$ using Black Box Variational Inference with reparameterization gradients (Kingma & Welling, 2014; Rezende et al., 2014). Since $p(\mathbb{S}|\lambda)$ is a discrete probability distribution, it is upper bounded by 1, so the optimization over $\lambda$ is well defined. As discussed above, this is in contrast to the loss $L$, which is not bounded as a function of $\lambda$.

We choose a fully factorized variational distribution $q_\gamma(\mathbf{E}, \mathbf{R})$. This is called the mean field approximation. Specifically, we use a fully factorized Gaussian variational distribution with means $\mu_{ei}^E$ and $\mu_{ri}^R$, and standard deviations $\sigma_{ei}^E$ and $\sigma_{ri}^R$ for the parameters $\mathbf{E}_{ei}$ and $\mathbf{R}_{ri}$, respectively. The collection of all means and standard deviations comprises the variational parameters $\gamma$.

**Learning Optimal Hyperparameters.** Algorithm 1 summarizes our method. It consists of three steps: (i) Pre-train a point-estimated model using the MAP estimator $L$ (Eqs. 2-4) with a reasonable choice of hyperparameters $\lambda$ (Line 3 in Algorithm 1). (ii) Use variational EM to maximize the lower bound on the marginal likelihood over the hyperparameters (Lines 4-9); here, we use the pretrained model to initialize the means of the variational distribution, see below. (iii) Plug the learned hyperparameters $\lambda$ into the MAP estimator and train the final model (Line 10).

Implementing these three steps requires only few changes compared to a MAP estimated model. Steps (i) and (iii) are just the existing MAP estimation of the latent parameters. The implementation of step 2 is also very similar. To optimize the ELBO with the reparameterization gradient trick, one injects parameter noise into the loss $L$ of the MAP estimator (see Lines 6-7). One then optimizes over the optimal amplitude $\sigma^{E/R}$ of the noise using stochastic gradient descent. The additional entropy term $-\log \sigma^E - \log \sigma^R$ in Line 7 prevents mode collapse of the variational distribution.

Lines 6-7 in Algorithm 1 describe the most generic way to estimate the gradient of the ELBO using reparameterization gradients. For specific models, parts of the ELBO can be evaluated analytically, which typically reduces the gradient noise. In the appendix, we derive an analytic expression for the contribution of the regularizer (log prior) to the ELBO. We find a solution in closed form for regularization with both the 2-norm and the 3-norm. The expected log prior prevents degenerate solutions of the kind discussed above because it penalizes sending $\lambda_{e/r}$ to infinity.

After maximizing ELBO over $\lambda$ and $\gamma$, we plug the learned $\lambda$ into the original MAP estimator and improve the performance of the model, see Line 10 in Algorithm 1. We also experimented with using the learned variational distribution $q_\gamma(\mathbf{E}, \mathbf{R})$ directly for link prediction using a Bayesian inference approach. In this setup, we evaluate the predictive probability of, e.g., a tail $t$ given head $h$ and relation $r$ by taking the expectation of the conditional probability $p(t|h, r, \mathbf{E}, \mathbf{R})$ under the variational distribution. We find that, when the embedding dimensions $K$ is small, these Bayesian predictions outperform point estimates. However, for a large embedding dimension $K$, MAP estimators perform better. We speculate that this may be because the match between variational distribution and true posterior may be worse in high dimensions, where the true posterior can have a more complicated structure that the mean field variational distribution cannot capture. We find that the main advantage of the variational treatment of the models that we considered is the efficient optimization over hyperparameters.

## 5 EXPERIMENTAL RESULTS

We test the performance of DistMult and ComplEx on four standard data sets, where we optimized hyperparameters in two different ways. Our baseline is standard grid search, and our proposed approach for hyperparameter tuning is variational EM (Algorithm 1). The latter relies on the probabilistic formulation provided in this paper. We considered two types of metrics: mean reciprocal rank, and hits at 10 (explained below). Our approach sets a new state-of-the-art in hits at 10 on all four data sets, and in mean reciprocal rank on three out of four data sets.

**Performance Metrics.** We report the standard metrics used in the knowledge graph embedding literature. We perform head and tail prediction and report all results in the 'filtered' setting introduced in (Bordes et al., 2013). In tail prediction, one tries to find the correct tail $t$ of a fact $(h, r, t)$ from the test set when given only head $h$ and relation $r$. We do this by ranking all candidate tails according to their probabilities under the model. The 'filtered' rank, denoted below by $\mathrm{rank}(t|h, r)$, takes into account that more than one tail may be a correct answer. One removes from the ranking all other tails $t' \neq t$ for which a fact $(h, r, t')$ exists in either the training, validation, or test set. We evaluate the following metrics of the filtered ranks (for all metrics, higher is better):

- **Mean Reciprocal Rank (MRR),** see, e.g., (Yang et al., 2015). It is defined by

$$\mathrm{MRR} = \frac{100}{\text{\# of facts in test set } \mathbb{S}'} \sum_{(h,r,t) \in \mathbb{S}'} \frac{1}{\mathrm{rank}(t|h, r)}. \tag{7}$$

- **Balanced Mean Reciprocal Rank (MRR$_b$).** As discussed in the introduction, standard data sets for knowledge graphs are highly imbalanced, i.e., most facts involve only a small subset of all entities. This biases the standard MRR metric. We therefore propose a new metric that is more

balanced than the standard MRR by averaging over entities rather than relations,

$$\text{MRR}_b = \frac{100}{N_e} \sum_{e \in [N_e]} \frac{1}{\text{\# of test triplets with tail } t} \left[ \sum_{\text{triplets with tail } t} \frac{1}{\text{rank}(t|h,r)} \right]. \quad (8)$$

- **Hits at 10 (H@10),** see, e.g., (Bordes et al., 2013). It is the percentage of test facts for which the 10 highest ranked predictions contain the correct prediction, i.e., for which $\text{rank}(t|h,r) \le 10$.

**Baselines.** We study two models, ComplEx (Trouillon et al., 2016; Lacroix et al., 2018) and Dist-Mult (Yang et al., 2015; Kadlec et al., 2017), with and without the variational EM algorithm. As in (Lacroix et al., 2018), our baseline for ComplEx, uses the weighted 3-norm regularizer that reached the previous state-of-the-art performance, i.e., $\lambda_{e/r}$ are proportional to the frequency of entity $e$ (relation $r$) in the dataset. We implemented our own version of DistMult using reciprocal relations and full log-loss with weighted 3-norm regularizer. The results were consistent with (and even slightly improved) the best results reported in the literature (Yang et al., 2015; Kadlec et al., 2017).

**Datasets.** We used four standard datasets. The first two are FB15K from the Freebase project (Bollacker et al., 2008) and WN18 from the WordNet database (Bordes et al., 2014). The other two datasets, FB15K-237 and WN18RR, are modified versions of FB15K and WN18 due to (Toutanova & Chen, 2015; Dettmers et al., 2018). The motivation for the modified datasets is that FB15K and WN18 contain near duplicate relations that make link prediction trivial for some facts, thus encouraging overfitting. In FB15K-237 and WN18RR these near duplicates were removed.

**Results.** Tables 1 and 2 summarize our results. We see that the balanced metric $\text{MRR}_b$ is generally lower than its unbalanced counterpart MRR, confirming the intuition that MRR is biased towards the easier predictions. Table 1 shows results for the ComplEx model. We see that variational EM improves the performance of the ComplEx model on three out of the four datasets. On the fourth dataset, FB15K, variational EM improves H@10 over the baseline, but MRR and $\text{MRR}_b$ are better in the baseline model. The latter may be explained by the fact that the FB15K data set is known to contain near duplicates in the test-training split, thus favoring models that tend to overfit the data.

For DistMult the improvements are more significant (Table 2). In WN18, variational EM improves both MRR and $\text{MRR}_b$ by 2 percentage points, and in WN18RR all metrics are improved almost by 1 percent. $\text{MRR}_b$ is improved by 0.5 percent in FB15K237, and H@10 is improved by 0.5 percent in FB15K. Although a more exhaustive hyper-parameter tuning using cross validation might further improve the performance of DistMult, the goal of this work is to reach such performance without a need to do an expensive grid search. Using variational EM hyperparameter learning algorithm, we can learn good hyperparameters $\lambda$ easily and reach the state-of-the-art results without spending many resources on hyperparameter tuning.

Table 1: Performance of different variants of the ComplEx model (filtered ranks). $^\dagger$ denotes our implementation of ComplEx with reciprocal relations and weighted 3-norm regularizer, hyperparameters taken from (Lacroix et al., 2018) (it reaches the same performance as (Lacroix et al., 2018)).

| Variant of the ComplEx model | WN18RR MRR/MRR$_b$/H@10 | WN18 "/"/" | FB15K237 "/"/" | FB15K "/"/" |
|---|---|---|---|---|
| (Trouillon et al., 2016) | -/-/- | 94.1/-/94.7 | -/-/- | 69.2/-/84.0 |
| Our MAP† | 48.2/47.0/57.2 | 95.2/**95.8**/96.2 | 36.4/19.9/55.6 | **85.8/82.0**/90.9 |
| Our EM | **48.6/47.3/57.9** | **95.3/95.8/96.4** | **36.5/20.3/56.0** | 85.4/81.9/**91.5** |

Table 2: Performance of different models for DistMult model (filtered). Bold numbers are the best performance. $^\dagger$Our implementation of DistMult with reciprocal relations and weighted 3-norm regularizer.

| Variant of the DistMult model | WN18RR MRR/MRR$_b$/H@10 | WN18 "/"/" | FB15K237 "/"/" | FB15K "/"/" |
|---|---|---|---|---|
| (Kadlec et al., 2017) | -/-/- | 79.0/-/95.0 | -/-/- | 83.7/-/90.4 |
| Our MAP† | 44.7/43.4/53.3 | 89.8/90.0/95.8 | 35.5/18.9/54.6 | **84.2**/80.0/90.8 |
| Our EM | **45.5/44.1/54.4** | **92.1/92.3/95.9** | **35.7/19.4/54.8** | 84.1/**80.2/91.4** |

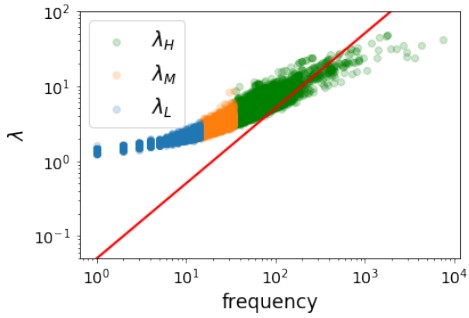

(a) Learned $\lambda$ for entities vs. frequencies.

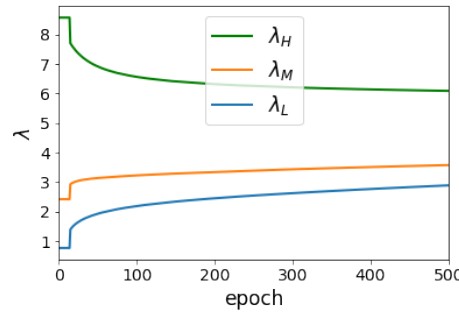

(b) Evolution of average $\lambda$ in each frequency bin.

Figure 3: The values of $\lambda$ at the end of optimization for FB15K-237 dataset and over training epochs. In Figure 3a, the red line is the weighted $\lambda$ in the baseline. Also, in Figure 3b $\lambda$ are initialized with the baseline. The plots justifies the heuristic of using a $\lambda$ proportional to the frequency of parameters. Figure 3a also confirms that there is a power-law relation between $\lambda$ and frequencies.

Finally, we study the influence of the frequency of entities in the training data on the hyperparameters learned by the variational EM algorithm. Figure 3a shows the learned $\lambda_e$ for all entities $e$ as a function of the entities' frequencies. The red line compares this to the weighted regularizer proposed in (Srebro & Salakhutdinov, 2010), which sets $\lambda_e$ and $\lambda_r$ proportional to the frequency. Qualitatively, our findings confirm the heuristic to use stronger regularization for entities with more training data. Quantitatively, however, we find that a linear relation between frequency and regularization strength is too strong. Our result might point to a better frequency based heuristics to choose $\lambda$.

To study the evolution of regularizations over the training time, we divide the entities into three bins $[L, M, H]$ of low, medium, and high frequency. Each bin contains one third of the entities. In Figure 3b shows the evolution of the average $\lambda$ within each bin as a function of the training epoch. The $\lambda$ are initialized with the baseline (i.e., $\lambda$ proportional to their frequency). We hold the hyperparameters fixed for the first 15 epochs of the EM algorithm to allow for the variational distribution $q_\gamma$ to come close to the actual posterior before we start updating $\lambda$. We see that on average, the hyperparameters learned by our approach converge to a different optimal value than the heuristic suggested by Srebro & Salakhutdinov (2010). In our experiments, we also see that, as expected, the inferred uncertainties $\sigma_e^E$ and $\sigma_r^R$ are higher for entities and relations with fewer training facts (see appendix for more details).

## 6  CONCLUSIONS

We showed that two of the most popular knowledge graph embedding models—DistMult and ComplEx—have an interpretation as probabilistic generative models of facts. Drawing on this view, we presented a scalable variational inference algorithm to learn a lower bound of the marginal likelihood of the data. We used this bound to optimize the many hyperparameters of these models by gradient descent (variational EM); an approach that would not be possible when point estimating model parameters. Using this methodology of optimizing hyperparameters, we outperformed the state of the art in link prediction on several popular benchmark data sets. The approach enjoys the same scalability properties as conventional maximum likelihood learning.

Our approach amounted to training the model twice: once in a Bayesian fashion to optimize hyperparameters, and then by point-estimating model parameters, using the hyperparameters obtained from the previous optimization. One may wonder why the approximate posterior was not used for link prediction. Our experiments revealed that using the posterior for link prediction worked well in low dimensions, but underperformed in high dimensions (not shown). We interpret this as a failure of the variational approximation in high dimension where the true posterior may locally not look Gaussian, pushing the variational Gaussian means away from the posterior maximum. For learning hyperparameters via variational EM, however, the variational approach works in all considered embedding dimensions. In the future, it would be interesting to explore more sophisticated posterior approximations that avoid this double-training procedure. However, a fully-Bayesian prediction task would be expensive to evaluate, making the proposed approach the most relevant one in practice.

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

# A APPENDIX

## A.1 COMPUTING THE TERMS IN ELBO

In this section , we compute the expectations appear in the ELBO in Eq. 6. For simplicity, we choose DistMult as the embedding model, but the computations are similar for the other models such as ComplEx (Trouillon et al., 2016) or CP (Hitchcock, 1927). The term $-\mathbb{E}_{q_\gamma}[\log q_\gamma(\mathbf{E}, \mathbf{R})]$ is the entropy of the Gaussian distribution, and it is

$$-\mathbb{E}_{q_\gamma}[\log q_\gamma(\mathbf{E}, \mathbf{R})] = \sum_{e \in [N_e]} \sum_{i \in [K]} \log \sigma_{ei}^E + \sum_{r \in [N_r]} \sum_{i \in [K]} \log \sigma_{ri}^R + c, \tag{9}$$

where $c$ is a constant and can be ignored in the optimization process. Now, let us focus on the evaluation of the expected prior. We derive the equations for $\mathbb{E}_{q_\gamma}[\log p(\mathbf{E}|\lambda_e)]$ and $\mathbb{E}_{q_\gamma}[\log p(\mathbf{R}|\lambda_r))]$ can be computed in a similar way.

**Gaussian prior.** First, let us focus on the Gaussian prior

$$p(\mathrm{E}_e|\lambda_e) = \sqrt{(\frac{\lambda_e}{2\pi})^K} \exp(-\lambda_e \|\mathrm{E}_e\|^2/2).$$

The log of Gaussian prior is

$$\log p(\mathrm{E}_e|\lambda_e) = -\frac{\lambda_e}{2}\|\mathrm{E}_e\|^2 + \frac{K}{2}\log \lambda_e + c. \tag{10}$$

The expectation of Eq. 10 with respect to the variational distribution $q_\gamma$ is

$$\mathbb{E}_{q_\gamma}[\log p(\mathrm{E}_e|\lambda_e)] = -\frac{\lambda_e}{2}\left(\|\mu_e^E\|^2 + \|\sigma_e^E\|^2\right) + \frac{K}{2}\log \lambda_e + c., \tag{11}$$

where $\|\sigma_e^E\|^2 = \sum_{i \in [K]}(\sigma_{ei}^E)^2$, (details can be found in Appendix B of (Kingma & Welling, 2014)). Notice that $\lambda_e$ only appears in $p(\mathrm{E}_e|\lambda_e)$, so for a given $\mu_e^E$ and $\sigma_e^E$ we can easily find its optimal value by maximizing $\mathbb{E}_{q_\gamma}[\log p(\mathrm{E}_e|\lambda_e)]$ over $\lambda$ as follows,

$$\lambda_e = \frac{K}{\|\mu_e^E\|^2 + \|\sigma_e^E\|^2}, \tag{12}$$

**Nuclear 3-norm prior.** Now, let us consider the prior

$$p(\mathrm{E}_e|\lambda_e) = \frac{1}{Z}\exp(-\frac{\lambda_e}{3}\|\mathrm{E}_e\|_3^3),$$

where $Z$ is the normalization factor. The normalization factor $Z$ is only a function of $\lambda_e$ and not the variational parameters $\gamma$, hence $\mathbb{E}_{q_\gamma}[\log Z] = \log Z$. We start the analysis by computing the normalization factor $Z$,

$$Z = \int_{\mathrm{E}_e \in R^K} \exp(-\frac{\lambda_e}{3}\|\mathrm{E}_e\|_3^3)\, d\mathrm{E}_e, \tag{13}$$

we compute the integral with the change of the variables $\zeta = \lambda_e^{1/3}\mathrm{E}_e$

$$Z = \frac{1}{\lambda_e^{K/3}}\int_{\zeta \in R^K} \exp(-\frac{1}{3}\|\zeta\|_3^3)\, d\zeta = \frac{c}{\lambda_e^{K/3}}, \tag{14}$$

where $c = \int_{\zeta \in R^K} \exp(-\frac{1}{3}\|\zeta\|_3^3)\, d\zeta$. Therefore,

$$\mathbb{E}_{q_\gamma}[\log Z] = \log Z = -\frac{K}{3}\log \lambda_e + c \tag{15}$$

Now, let us focus on computing

$$\mathbb{E}_{q_\gamma}[\frac{\lambda_e}{3}\|\mathrm{E}_e\|_3^3] = \mathbb{E}_{q_\gamma}[\frac{\lambda_e}{3}\sum_{i \in [K]}|\mathrm{E}_{ei}|^3].$$

Given the Gaussian variational distribution $N(\mu_{ei}^E, (\sigma_{ei}^E)^2)$, the samples of $N(\mu_{ei}^E, (\sigma_{ei}^E)^2)$ can be written as

$$\mathrm{E}_{ei} \sim \mu_{ei}^E + \sigma_{ei}^E \epsilon, \tag{16}$$

where $\epsilon \in \mathbb{R}$ is the noise sampled from the Gaussian distribution $N(0,1)$. Using this reparametrization, we get

$$
\begin{aligned}
\mathbb{E}_{q_\gamma}\left[\|\mathrm{E}_{ei}\|_3^3\right] &= \frac{1}{\sqrt{2\pi}} \int_{-\infty}^{\infty} |\mu_{ei}^E + \sigma_{ei} x|^3 e^{-\frac{1}{2}x^2}\, dx \\
&\overset{\mu_0 = \frac{\mu_{ei}^E}{\sigma_{ei}}}{=} \frac{\sigma_{ei}^3}{\sqrt{2\pi}} \int_{-\infty}^{\infty} |\mu_0 + x|^3 e^{-\frac{1}{2}x^2}\, dx \\
&= \frac{\sigma_{ei}^3}{\sqrt{2\pi}} \left[ -\int_{\mu_0}^{\infty} (-\mu_0 + x)^3 e^{-\frac{1}{2}x^2}\, dx + \int_{-\mu_0}^{\infty} (\mu_0 + x)^3 e^{-\frac{1}{2}x^2}\, dx \right] \\
&= \frac{\sigma_{ei}^3}{\sqrt{2\pi}} \left[ 2\int_{\mu_0}^{\infty} (3\mu_0^2 x + x^3) e^{-\frac{1}{2}x^2}\, dx + \int_{-\mu_0}^{\mu_0} (3\mu_0 x^2 + \mu_0^3) e^{-\frac{1}{2}x^2}\, dx \right] \\
&= \frac{\sigma_{ei}^3}{\sqrt{2\pi}} \left[ 2(\mu_0^2 + 2)e^{-\frac{1}{2}\mu_0^2} + \sqrt{\frac{\pi}{2}} \left(6\mu_0 + 2\mu_0^3\right) \operatorname{erf}(\frac{\mu_0}{\sqrt{2}}) \right] \\
&= \sqrt{\frac{2}{\pi}} \left((\mu_{ei}^E)^2 \sigma_{ei} + 2(\sigma_{ei})^3\right) e^{-\frac{1}{2}\left(\frac{\mu_{ei}^E}{\sigma_{ei}}\right)^2} + \left(3|\mu_{ei}^E|(\sigma_{ei})^2 + |\mu_{ei}^E|^3\right) \operatorname{erf}\left(\frac{|\mu_{ei}^E|}{\sqrt{2}\sigma_{ei}}\right),
\end{aligned}
\tag{17}
$$

where $\operatorname{erf}(\cdot)$ is the error function. Given $\mathbb{E}_{q_\gamma}\left[\|\mathrm{E}_e\|_3^3\right]$ we can compute the optimal $\lambda$ for a given $\gamma$ by maximizing ELBO as

$$\lambda_e = \frac{K}{\mathbb{E}_{q_\gamma}\left[\|\mathrm{E}_e\|_3^3\right]}. \tag{18}$$

**Arbitrary prior.** For an arbitrary prior, we can always use the reparameterization trick to compute a stochastic estimate of log prior and its gradient (check Lines 4-6 in Algorithm 1).

## A.2 Further Remarks

**Remark 1.** DistMult is further improved by Trouillon et al. (2016). The improved algorithm (ComplEx) embeds the entities and relations to a $K$ dimensional complex space $\mathbb{C}^K$ rather than $\mathbb{R}^K$, and $X_{h,r,t} = \operatorname{Re}\left(\sum_{i=1}^K \mathrm{E}_{hi}\mathrm{R}_{ri}\bar{\mathrm{E}}_{ti}\right)$. Using complex numbers increases the flexibility of the algorithm and allows ComplEx to capture the non-symmetry in the triplet facts, meaning that $X_{h,r,t} \neq X_{t,r,h}$. This happens when the imaginary parts of the entries of the $\mathrm{R}_r$ are non-zero, while if the imaginary parts of the entries of the $\mathrm{R}_r$ are zero, the model can learn the symmetric relations, i.e., $X_{h,r,t} = X_{t,r,h}$. This is in contrast to DistMult, where always $X_{h,r,t} = X_{t,r,h}$.

**Remark 2.** Lacroix et al. (2018) presented a data augmentation technique, which simplifies the presentation of the model. Given the set of the triplet facts $\{(h_i, r_i, t_i)\}_{i=1,\ldots,n}$, one first augments the data set by adding a new 'reciprocal' fact for each existing fact $(h_i, r_i, t_i)$. Then, for each fact $(h_i, r_i, t_i)$ in the original data set, one adds a new fact by swapping head $h_i$ and tail $t_i$ and replacing the relation with its reciprocal relation. The new dataset $\mathbb{S} = \{(h_i, r_i, t_i)\}_{i=1,\ldots,2n}$ has $2n$ facts, $N_e$ entities and $2N_r$ relations ($N_r$ relations plus $N_r$ reciprocal relations). This augmentation of the data maps the head and tail prediction task to just tail prediction. Two advantages are: (i) better performance, this is because the relations are trained specifically for the task of tail predictions rather than training a relation for both head and tail predictions (Lacroix et al., 2018); and (ii) it establishes a consistent causal direction $(h, r) \to t$ that enables us to reuse the generative model for both head and tail predictions.

## A.3 Uncertainty Analysis

Figure 4 depicts the average standard deviation (i.e., $\sigma^E$ and $\sigma^R$) of the embeddings inferred by variational EM algorithm. As it is seen, on average, the frequent entities or relations have a lower

Table 3: Dataset Statistics.

| Dataset | #entities | #relations | #training$\times 10^3$ | #test$\times 10^3$ | # validation$\times 10^3$ |
|---|---|---|---|---|---|
| FB15K237 | 15k | 237 | 272k | 20k | 18k |
| FB15K | 15K | 1k | 500k | 60k | 50k |
| WN18RR | 41k | 11 | 87k | 3k | 3k |
| WN18 | 41k | 18 | 141k | 5k | 5k |

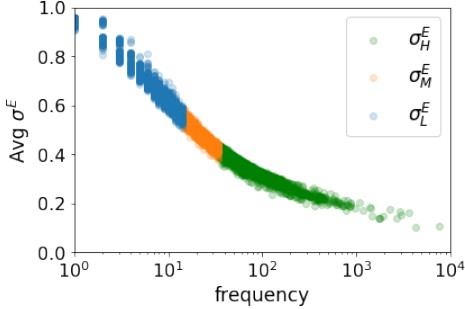

(a) Uncertainty of entities' embeddings vs frequencies.

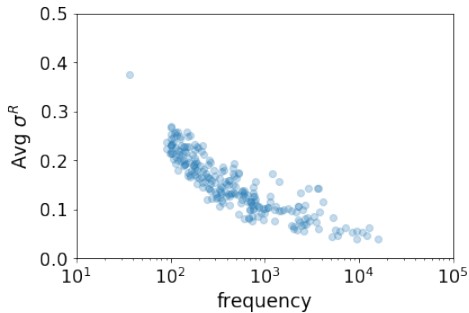

(b) Uncertainty of relations' embeddings vs frequencies.

Figure 4: The average of standard deviations of Gaussian variational distribution for FB15K237 in variational EM algorithm. We see that on average the entities with higher frequencies reach a lower $\sigma$, that means a lower uncertainty.

uncertainty and variational EM algorithm learns more confident embeddings for the frequent entities or relations. Also, on average, the relations' embeddings have lower uncertainty compared to the embeddings' uncertainty. This is because there are fewer relations compared to entities, hence on average, there are more facts for a relation compared to an entity.

