# OpenReview forum: "Probabilistic Knowledge Graph Embeddings"
_ICLR.cc/2019/Conference_

### Official Review · AnonReviewer1 · 2018-11-02
**An adaptive hyperparameter tuning method for knowledge graph embeddings**

**Rating:** 5
**Confidence:** 3

**Review:**

This paper proposes a Bayesian extension to knowledge base embedding methods, which can be used for hyperparameter learning. My rating is based on following aspects.

Novelty.
Applying Bayesian treatment to embedding methods for uncertainty modelling and hyperparameter tuning is not new (examples include PMF [1] and Bayesian PMF [2]), and Sec 3 can be regarded as a knowledge base extension of them with a different likelihood (MF considers user-item pairs while knowledge base considers head-edge-tile triplets). However, it seems that there is little work considering the hyperparameter tuning problems for knowledge base embeddings.

Quality & Clarity.
This paper makes two arguments. 1. Small data problems exist, and needs parameter uncertainty; 2. Bayesian treatment allows efficient optimization over hyperparameters. However, as mentioned in Sec 4 and Sec 5, they still use MAP estimations with tuned hyperparameters instead of variational distribution directly. This does not support the parameter uncertainty argument (since there is no uncertainty in parameters of the final model, i.e., those re-trained in line 10 of algorithm 1). More analysis, both theoretically and experimentally, is needed to address this argument. The hyperparameter tuning argument is well-supported by both theoretical analysis and experiments.

My questions are mainly about experiments. Overally, I think current experiments cannot support the claims well and further experiments are needed.
1.	As mentioned above, the parameter uncertainty issue hasn’t been well verified (Figure 3 demonstrates the advantages of hyperparameter tunning instead of uncertainty in parameters).
2.	Table 1 & 2 demonstrates that hyperparameter tunning using algorithm 1 introduces performance improvement on ComplEx and DistMult. Since the Bayesian treatment is general, such an improvement should be found for other knowledge base embedding methods.
3.	Time complexity is not analyzed (since Algorithm 1 requires re-train the models).
4.	Algorithm 1 is a one-EM-step approximation for optimizing the ELBO. How well such a algorithm approximates the optimal solution of ELBO. For example, what will happens if running line 4-10 for multiple times? Does the performance increase or decrease?

[1] Salakhutdinov and Minh, Probabilistic Matrix Factorization, NIPS 2007.
[2] Salakhutdinov and Minh, Bayesian Probabilistic Matrix Factorization using Markov Chain Monte Carlo, ICML 2008.

---

> ### Author Response · Authors · 2018-11-20
> **Thank you for your detailed comments!**
>
> We respond to each of your questions below:
>
> > Sec 3 can be regarded as a knowledge base extension of [Bayesian treatments to
> > embedding methods] with a different likelihood [...].
>
> We agree with the reviewer’s summary of our theoretical contributions. In addition, we would like to stress our experimental contribution. Our method sets a new state of the art on a very competitive benchmark, while at the same time significantly reducing the cost for hyperparameter tuning.
>
> > [...] However, [...] they still use MAP estimations with tuned hyperparameters
> > instead of variational distribution directly. This does not support the
> > parameter uncertainty argument [...]. The hyperparameter tuning argument is
> > well-supported by both theoretical analysis and experiments.
>
> We will clarify the role of uncertainty. The final prediction does indeed not take uncertainty into account, as we discuss in the conclusions. The proposed hyperparameter optimization algorithm, however, only works because of a Bayesian treatment of the latent embedding vectors. As we mention in the paragraph above the algorithm box, a gradient based hyperparameter optimization that ignores posterior uncertainty would lead to divergent solutions.
>
> Specifically, minimizing the loss function $L$ simultaneously over model parameters and hyperparameters would send the hyperparameters to infinity. This minimizes the loss if the model parameters are strictly zero, which is possible in a point estimated model but not when we attribute a nonzero uncertainty to each parameter.
>
> > [...] the parameter uncertainty issue hasn’t been well verified (Figure 3
> > demonstrates the advantages of hyperparameter tuning instead of uncertainty
> > in parameters).
>
> Due to space limitations, we defer the quantification of uncertainty to the appendix. Figure 4 shows the posterior uncertainty as a function of the frequency of an entity (or relation). It shows a clear correlation between infrequent entities and high posterior uncertainty.
>
> > Since the Bayesian treatment is general, such an improvement [as shown for
> > the ComplEx and DistMult models] should [also] be found for other knowledge
> > base embedding methods.
>
> In our experiments, we focused on the DistMult model because of its simplicity, and on the ComplEx model because it is the current state of the art and because it is a particular hard benchmark for a new hyperparameter tuning method as a lot of effort has already been invested into tuning its hyperparameters, see [Kadlec et al., 2017] and [Lacroix et al., 2018].
>
> We agree that the proposed method is more general and that it should be easily applicable to other knowledge graph embedding models. We think that the main advantage of our method will be to speed up the evaluation cycle when designing new models. Current records on the link prediction task are held by tensor factorization models that may seem surprisingly primitive given the difficulty of the task. Yet, without our method, it would be difficult to prove that a more involved model performs better in practice because one would have to compete with the large amount of expensive hyperparameter tuning that has already gone into existing models.
>
> > Time complexity is not analyzed (since Algorithm 1 requires re-train the models).
>
> Our method scales linearly in the number of hyperparameters. By comparison, the standard approach to hyperparameter optimization, i.e., grid search using a validation set, scales exponentially with the number of hyperparameters. It would be infeasible to perform grid search over individual hyperparameters for each entity and relation.
>
> When comparing our method to training without hyperparameter optimization, we obtain the following: the first and the last of the three training cycles in our method are the usual MAP estimation of the parameters. The second training cycle (variational EM) took less time than these two steps (due to the preinitialization).
>
> > Algorithm 1 is a one-EM-step approximation for optimizing the ELBO. [...]
> > For example, what will happens if running line 4-10 for multiple times?
>
> Running lines 4-10 multiple times would mean that we would alternate between variational EM (lines 4-9) and re-training with the learned hyperparameters (line 10). This would be wasteful since both parts of the algorithm optimize over the model parameters, just under different approximations (variational approximation for lines 4-9 vs. point estimate approximation for line 10). Returning to variational EM (lines 4-9) after executing line 10 would cause the algorithm to forget anything that it has learned on line 10. By contrast, the opposite direction preserves information. Lines 4-9 also optimize over the hyperparameters $\lambda$, which are then used (and left unchanged) on line 10. This is why it is important to do variational EM (lines 4-9) first, before one re-trains with the learned hyperparameters (line 10).

---

### Official Review · AnonReviewer3 · 2018-11-02
**Sound method, but the scope is limited to hyperparameter tuning, and no comparisons with other methods are provided**

**Rating:** 6
**Confidence:** 3

**Review:**

In this paper, authors propose a probabilistic extension of classic Neural Link Prediction models, such as DistMult and ComplEx. The underlying assumption is that the entity embeddings and the relation embeddings are sampled from a prior Multivariate Normal distribution, whose (hyper-)parameters can be estimated via maximum likelihood. In this paper, authors use Variational Inference (VI) for approximating the posterior distribution over the embeddings, and use Stochastic VI for maximising the Evidence Lower BOund (ELBO) while scaling to large datasets. In Sect. 3, authors introduce the generative process, and show how MAP estimation of the embedding matrices can recover the original models. In Sect. 4, authors start from the intractable marginal likelihood over the data (Eq. 5) for deriving the corresponding ELBO (Eq. 6), which is defined over:
- The "hyperparameters" gamma, which define the parameters of the prior Multivariate Normal distribution over the embeddings, and
- The parameters gamma of the variational distributions.

Question: why the ELBO (Eq. 6) is not used anywhere in Algorithm 1?

The model does not mention a number of significantly more accurate models proposed in the literature, such as [1].

Furthermore, it seems to me that the point of the whole paper is finding efficient ways of estimating the hyperparameters efficiently. In that sense, there are other methods that were not considered, either simple (e.g. random sampling or black-box optimization techniques [2]) or more complex (e.g. hypergradient descent [3]).

[1] https://arxiv.org/abs/1707.01476
[2] https://github.com/hyperopt/hyperopt
[3] https://arxiv.org/abs/1703.04782

---

> ### Author Response · Authors · 2018-11-20
> **Thank you for your valuable feedback**
>
> We respond to each question below:
>
> > Question: why is the ELBO (Eq. 6) not used anywhere in Algorithm 1?
>
> Thank you for this question. In fact, line 7 in Algorithm 1 does contain an unbiased estimator of the ELBO (we will clarify this notation). We decided to write out the ELBO in terms of the loss function L in the algorithm box so as to stress that it is easy to retrofit the EM algorithm into an existing implementation of a model that just minimizes a loss L.
>
> In detail, the ELBO (Eq. 6) is given by the expected log joint distribution minus the expected log variational distribution. The latter can be calculated analytically and leads to the $\log \sigma$ terms on line 7 (the entropy of q). We obtain an unbiased estimator of the (negative) expected log joint by injecting Gaussian noise with standard deviation $\sigma$ into the loss function L.
>
> > The model does not mention a number of significantly more accurate models
> > proposed in the literature, such as [1].
> > [1] https://arxiv.org/abs/1707.01476
>
> We present comparisons of our experimental results to the best baseline that we could find in the literature. The mentioned paper [1] reports results on the same four benchmark datasets that we also use (compare Tables 3 and 4 in [1] to Tables 1 and 2 in our paper). Only for one dataset (WN18), results reported in [1] are comparable to the baselines used in our paper. On all three other datasets, the baselines used in our paper perform substantially better (>5 percentage point improvement for Hits@10).
>
> > Furthermore, it seems to me that the point of the whole paper is finding
> > efficient ways of estimating the hyperparameters efficiently. In that sense,
> > [...] there are other methods [to estimating hyperparameters efficiently]
> > that were not considered [...]
> > [2] https://github.com/hyperopt/hyperopt
> > [3] https://arxiv.org/abs/1703.04782
>
> Thank you for pointing out these references. The library in [2] is a general purpose optimization library that takes an objective function and tries to find its maximum. It is specialized for searches over parameter spaces of nontrivial shape. This is not necessary in our hyperparameter optimization problem since the search space is a simple real-valued vector space. The documentation for [2] suggests that the library may in the future support Bayesian optimization. Bayesian optimization is, in principle, an alternative method for hyperparameter optimization, but it does not scale well to a large number of hyperparameters as it requires retraining the full model after every change to any hyperparameter. In contrast, variational EM optimizes hyperparameters and model parameters concurrently.
>
> The work by [3] is orthogonal to our work. It proposes a learning algorithm for the hyperparameters of the optimizer (e.g., the learning rate) to improve the convergence rate of gradient descent. In contrast, our contribution optimizes over hyperparameters of the model, such as regularizers.

---

### Official Review · AnonReviewer2 · 2018-11-09
**Probabilistic Knowledge Graph Embeddings - Review**

**Rating:** 5
**Confidence:** 2

**Review:**

Summary:
The paper presents a probabilistic treatment of knowledge graph embeddings, motivating it in parameter uncertainty estimation and easier hyperparameter optimisation. The authors present density-based DistMult and ComplEx variants, where the posterior parameter distributions for entity and relation embeddings are approximated by diagonal Gaussians q_\gamma. Variational EM is used to infer the variational parameters \gamma as well as the per-entity/per-relation precision (\lambda) hyperparameters. The training process proposed by the authors consists of three phases: (1) pretraining a MAP estimate that’s used as initial means of the posterior approximating Gaussians, (2) variational EM (see above) to find better hyperparameters and (3) another MAP training phase that uses the updated per-entity/per-relation hyperparameters. Finally, experimental results indicate a slight improvement in MRR and HITS@10 across FB and WN datasets.

Originality:
To the best of the reviewer’s knowledge, the presented approach is novel for knowledge graph embeddings.

Discussion:
While the task is relevant, it is unclear how significant the improvements are. While overall, the proposed method seems to indicate small improvement upon a very strong baseline, in some cases it’s very close (96.2 vs 96.4 HITS@10 on WN18, 36.4 vs 36
.5 MRR on FB15K237), or worse (85.8 vs 85.4 MRR on FB15K).
It is unclear how adequate some details in the experimental setup are for verifying the main hyperparameter optimization claim. In particular, what is “a reasonable choice of hyperparameters” in the first training phase? From figure 3b it seems the initial lambda’s are set proportionately to the frequency, as in the baseline. Are the initial hyperparameter values in EM set the same as the hyperparameter values used for MAP in the reported results? If the claim is to optimize hyperparameters, shouldn’t their initial values be set as uninformed as possible? How do different initial hyperparameter values affect final performance?
The authors claim that the improvement is most notable for entities with fewer training points, however, this is only investigated by using a balanced MRR, where the results are again very close, the same (WN18) or worse (FB15K) for ComplEx. Wouldn’t it be clearer to perform a separate evaluation only considering low-frequency entities to verify this claim?
Parameter uncertainty is not further handled in the paper, the final approach is a point estimate, which discards the uncertainties obtained by VI. Authors mention (last paragraph of Sec. 4) that for a large embedding dimension, bayesian predictions are worse, while for small dimension, they are better. The author’s hypothesis is that a more flexible posterior approximation could solve this issue. No concrete numbers or further analysis are provided.


Clarity and presentation:
The result tables should be merged and formatted better.
Figures need some work (Fig. 2 looks poorly scaled, all figures should be in vector format for scalability, typos in Fig. 1)

Questions:
- How much additional computation is needed to achieve the reported results?
- Would it be possible to group the entities in bins by frequencies (say 6-10 bins) and assign each bin a hyperparameter, and run grid search over just 6-10 hyperparameters, and then interpolate between the bins to set hyperparameters per entity as a function of its frequency?

---

> ### Author Response · Authors · 2018-11-20
> **Thank you very much for your insightful comments**
>
> We respond to each of your questions below:
>
> > [...] While overall, the proposed method seems to indicate small improvement
> > upon a very strong baseline, in some cases it’s very close [...]
>
> Thank you for raising the point of a challenging baseline comparison. While the comparison is arguably close for the ComplEx model, improvements on the DistMult model are much more pronounced. Also, note that our proposed hyperparameter optimization method is much more efficient than the traditional grid search approach. We obtained our results from a single run of our method, whereas the baseline relied on the extensive hyperparameter search reported in [Lacroix et al., 2018]. We hope that our fast hyperparameter tuning method will speed up research on new knowledge base embedding models.
>
> > How do different initial hyperparameter values affect final performance?
>
> The choice of hyperparameters in the first training phase (the "pre-training") is only used to find good initializations for the embedding vectors in the second phase (the variational EM). We ran experiments both with uniform initial $\lambda$ and with initial $\lambda$ proportional to the frequency and obtained indistinguishable performance at the end.
>
> > The authors claim that the improvement is most notable for entities with fewer
> > training points, however, this is only investigated by using a balanced MRR [...]
>
> Thank you for pointing this out. We will revise the paper with the following more carefully formulated claim: the role of uncertainty is most important for entities with few training points. This is confirmed by Figure 4 in the appendix, which plots the posterior uncertainty as a function of frequency.
>
> > Parameter uncertainty is not further handled in the paper, the final approach
> > is a point estimate [...].
>
> We will clarify the role of uncertainty in the proposed hyperparameter optimization strategy. The final prediction does indeed not take uncertainty into account, as we discuss in the conclusions. The proposed hyperparameter optimization algorithm, however, only works because of a Bayesian treatment of the latent embedding vectors. As we mention in the paragraph above the algorithm box, a gradient based hyperparameter optimization that ignores posterior uncertainty would lead to divergent solutions.
>
> Specifically, minimizing the loss function $L$ simultaneously over model parameters and hyperparameters would send the hyperparameters to infinity. This minimizes the loss if the model parameters (=embedding vectors) are strictly zero, which is possible in a point estimated model but not in a model that attributes a nonzero uncertainty to each embedding vector. Technically, the optimization cannot diverge because the ELBO is bounded, see discussion below Eq. 6.
>
> > The author’s hypothesis is that a more flexible posterior approximation could
> > solve this issue. No concrete numbers or further analysis are provided.
>
> We will revise the conclusions and provide the following more concrete proposals. A more flexible posterior approximation opens up many new avenues of research. One option is to explicitly introduce a more flexible posterior approximation via a model based variational distribution. The challenge here is a trade-off between efficient inference and expressiveness of the variational model. Alternatively, the bound on the marginal likelihood can be improved by importance weighting, similar to [https://arxiv.org/abs/1808.09034]. This implicitly fits a more involved variational distribution (see Theorem 1 in that preprint).
>
> > How much additional computation is needed to achieve the reported results?
>
> Our approach is much cheaper than the standard approach for hyperparameter optimization, i.e., grid search using a validation set. The computational cost of grid search scales exponentially with the number of hyperparameters, and it would be infeasible to perform grid search over individual hyperparameters for each entity and relation. Our proposed method scales only linearly in the number of hyperparameters.
>
> When comparing our method to training without hyperparameter optimization, we obtain the following: the first and the last training cycles in our method are the usual MAP estimation of the parameters. The second training cycle (variational EM) took less time than these two steps (due to the preinitialization).
>
> > Would it be possible to group the entities in bins by frequencies
> > (say 6-10 bins) [...] and run grid search over just 6-10 hyperparameters [...]?
>
> It is possible, however, with two caveats. First, the optimal hyperparameter is not a strict function of the frequency, as can be seen by the spread in vertical direction of the data points in Figure 3a (note the log scale). Second, as grid search scales exponentially in the number of hyperparameters, even a grid search over only 6-10 hyperparameters would be much more expensive than our method, which involves only three training cycles.

---

### Meta-Review · Area_Chair1 · 2018-12-14
**Intersting idea but significance is not fully clear**

**Confidence:** 4
**Recommendation:** Reject

**Metareview:**

The paper proposes a Bayesian extension to existing knowledge base embedding methods (like DistMult and ComplEx), which is applied for for hyperparameter learning. While using Bayesian inference for for hyperparameter tuning for embedding methods is not generally novel, it has not been used in the context of knowledge graph modelling before. The paper could be strengthened by comparing the method to other strategies of hyperparameter selection to prove the significance of the advantage brought by the method.